# Factors Associated with Recurrent Heart Failure during Incorporating SGLT2 Inhibitors in Patients Hospitalized for Acute Decompensated Heart Failure

**DOI:** 10.3390/jcm11175027

**Published:** 2022-08-26

**Authors:** Masaki Nakagaito, Teruhiko Imamura, Shuji Joho, Ryuichi Ushijima, Makiko Nakamura, Koichiro Kinugawa

**Affiliations:** Second Department of Internal Medicine, University of Toyama, Toyama 930-0194, Japan

**Keywords:** heart failure, renal function, diabetes mellitus, ejection fraction

## Abstract

Background: Sodium-glucose cotransporter 2 inhibitors (SGLT2i) reduce the risk of hospitalization for heart failure (HF) or death from cardiovascular causes among patients with chronic HF. However, little is known about the specific factors associated with clinical events during SGLT2i therapy in patients hospitalized for acute decompensated heart failure (ADHF). Methods: Consecutive patients who were hospitalized for ADHF and received SGLT2i during the index hospitalization between February 2016 and April 2021 were retrospectively evaluated. We investigated the factors associated with recurrent hospitalization for HF during the SGLT2i therapy. Results: A total of 143 patients (median age 73 years, 92 men) were included. Estimated glomerular filtration rate (eGFR) was negatively associated with a primary endpoint with a hazard ratio of 0.94 (95% confidence interval 0.90–0.98, *p* = 0.007). Those with lower eGFR < 40.9 mL/min/1.73 m^2^ (*n* = 47) had significantly lower freedom from HF hospitalization during 1-year therapeutic period (73% versus 94%, *p* = 0.005). Conclusions: Among patients who initiated medical therapy incorporating SGLT2i during the hospitalization for ADHF, a lower eGFR at baseline was associated with a recurrent hospitalization for HF. Early administration of SGLT2i prior to deterioration of renal function would be highly recommended to enjoy greater benefit from SGLT2i.

## 1. Introduction

To date, sodium-glucose cotransporter 2 inhibitors (SGLT2i) represented a major therapeutic advance in patients with heart failure (HF). Large-scale clinical trials have shown that they reduced the risk of hospitalization for HF in patients with stable chronic HF with reduced left ventricular ejection fraction (HFrEF) [1,2]. Empagliflozin, one of the SGLT2i, additionally reduced the risk of HF hospitalization in patients with chronic HF with a preserved left ventricular ejection fraction (HFpEF) [3]. Furthermore, these beneficial effects were demonstrated irrespective of the existence of diabetes mellitus (DM). Thus, SGLT2i has become one of the essential therapeutic agents for chronic HF thus far.

Acute decompensated heart failure (ADHF) is a common cause of hospitalization in elderly people and is associated with a high postdischarge mortality and rehospitalization risk [4,5]. Prevention of recurrent hospitalization for HF is essential for improving cardiovascular outcome for them.

Recently, a large-scale trial demonstrated that sotagliflozin, initiated before or shortly after discharge, resulted in a significantly lower total number of recurrent HF hospitalization than placebo in patients with DM and recent worsening HF [6]. However, it remains unclear whether the effectiveness of SGLT2i varies depending on the background factors in patients hospitalized for ADHF. Thus, we investigated the factors associated with recurrent hospitalization for HF during the SGLT2i-incorporated medical therapy among patients hospitalized for ADHF.

## 2. Materials and Methods

The present study was a single-center, retrospective observational study designed to investigate the baseline factors associated with HF recurrence during SGLT2i-incorporated medical therapy among the patients hospitalized for ADHF. The local Institutional Ethics Board approved the study protocol, which complied with the Declaration of Helsinki. Written informed consent was obtained from all of the patients beforehand.

### 2.1. Study Population

This study involved patients who had been admitted for ADHF, which was diagnosed according to the Framingham criteria, at our institute between February 2016 and April 2021. All patients had New York Heart Association (NYHA) class III/IV symptoms upon admission. Participants received guideline-directed medical therapy for HF, including renin-angiotensin system inhibitors, beta-blockers, mineralocorticoid receptor antagonists, and diuretics, if applicable.

We included patients newly receiving SGLT2i during their index hospitalization immediately following the stabilization of hemodynamics. We defined DM as satisfying glycated hemoglobin (HbA1c) ≥ 6.5% or receiving antidiabetic treatment.

Exclusion criteria were as follows: age < 20 years, end-stage renal failure with estimated glomerular filtration rate (eGFR) < 15 mL/min/1.73 m^2^, use of durable left ventricular assist devices, pregnancy or breastfeeding, and current use of SGLT2i on the index hospitalization. These patients were not included in this study. Patients who discontinued SGLT2i during the index hospitalization were also excluded.

### 2.2. Study Design and Data Collection

Baseline characteristics including demographics and laboratory data at index discharge were retrieved. The eGFR was calculated using the guidelines from the Chronic Kidney Disease Epidemiology Collaboration. Plasma B type natriuretic peptide (BNP) level was retrospectively retrieved at 3-, 6-, and 12-month follow-ups. Standard echocardiographic findings were retrieved during index hospitalization. For the present analysis, participants were divided into HFrEF (LVEF [left ventricular ejection fraction] < 40%), HFmrEF (LVEF 40–49%), and HFpEF (LVEF ≥ 50%).

Adjustment of medical therapy, except for SGLT2i, was permitted as real-world clinical practice. Discontinuation of SGLT2i was not permitted during the observational period in principle. When discontinued, the follow-up was terminated at that time. When patients died without being hospitalized for HF, they were censored at the time of events.

The primary outcome was an unplanned recurrent hospitalization for HF during the 12 months following their index discharge. The secondary outcome was the changes in plasma BNP level during the observational period. Baseline characteristics that would significantly affect these outcomes were investigated.

### 2.3. Statistical Analyses

The statistical analysis was performed by using JMP^®^ 15 (SAS Institute Inc., Cary, NC, USA). The level of significance was defined as 2-tailed *p* < 0.050. Continuous variables were expressed as the median and interquartile unless there were. any specific statements. Categorical variables were expressed as absolute numbers and percentages. A Wilcoxon test was applied to compare continuous parameters, and Pearson’s χ^2^ test was applied for comparison of categorical variables. Kruskal–Wallis test was applied to compare three continuous parameters. Trends on continuous variables were compared using a Friedman test.

Time-to-event data were evaluated using Kaplan–Meier estimates and Cox proportional hazards models to investigate the impact of baseline variables on clinical outcomes. Univariable and multivariable analyses with Cox proportional hazards models were performed to calculate the adjusted hazard ratio to assess the influence of various parameters on primary endpoint. Variables significant with *p* < 0.050 in the univariable analyses were included in the multivariable analyses. A cut off of eGFR for primary endpoint was calculated using receiver operating characteristic analysis with a logistic regression model.

## 3. Results

### 3.1. Follow-Up and Patient Characteristics

From February 2016 to April 2021, 206 patients initiated SGLT2i during their hospitalization for ADHF. Of them, 42 patients did not continue SGLT2i, 4 died during hospitalization, 1 underwent implantation of left ventricular assist device, and 16 patients were lost to follow-up after index discharge. Finally, a total of 143 patients were included in this study (Figure 1).

Early discontinuation of this study occurred in 28 patients (19.6%) due to the following reasons: 8 due to adverse events (symptomatic hypotension, diarrhea, and suspected bacterial infection), 2 due to death from noncardiovascular cause, 14 due to lost to follow-up, and 4 due to any other reasons.

Table 1 lists the baseline characteristics. The median age was 73 (65–81) years and 36% were women. HFrEF was noted in 53 patients (37%). DM was noted in 117 patients (82%). All patients with DM were those with type 2 DM. Baseline HbA1c level was 6.8% (6.5–7.7%). Baseline plasma BNP level was 142 (63–316) pg/mL.

### 3.2. Primary Outcome

As a primary outcome, a total of 15 patients (10.5%) had an unplanned recurrent hospitalization for HF. Univariate and multivariate analyses demonstrated that eGFR (hazard ratio 0.94, 95% confidence interval 0.90–0.98) was independently and negatively associated with the rehospitalization for HF (*p* = 0.007; Table 2).

A cut-off of baseline eGFR to predict the primary endpoint was 40.9 mL/min/1.73 m^2^ (0.744 of area under the curve). A total of 47 patients had eGFR < 40.9 mL/min/1.73 m^2^ (lower eGFR group) and the other 96 had eGFR ≥ 40.9 mL/min/1.73 m^2^ (higher eGFR group). The primary outcome was encountered more frequently in the lower eGFR group than in the higher eGFR group (*p* = 0.005; Figure 2).

Furthermore, there was significant difference in the incidence rates for the recurrent HF hospitalization among three EF groups (*p* = 0.040; Figure 3). Notably, the incidence of primary outcome was somewhat higher in HFrEF than in HFmrEF and HFpEF (*p* = 0.052 and *p* = 0.051, respectively). There was no significant difference in the risk of rehospitalization for HF between patients with DM and those without DM (*p* = 0.542, Figure 4).

### 3.3. Secondary Outcome

The changes of plasma BNP level were evaluated in 70 patients, excluding those who prematurely discontinued SGLT2i or those for whom BNP was not retrieved. The changes of plasma BNP level from baseline are presented in Figure 5. The higher eGFR group, HFrEF, and patients with DM had a significant absolute improvement in plasma BNP from baseline to 12 months. On the other hand, the lower eGFR group, HFmrEF, HFpEF, and patients without DM had no significant change in BNP throughout the 12-month observation period.

## 4. Discussion

We investigated the factors associated with HF recurrence during medical therapy incorporating SGLT2i, which was initiated during the index hospitalization for ADHF. A lower eGFR was associated with recurrent HF hospitalization. HFrEF trended to be associated with HF rehospitalization. HF rehospitalization was not different between those with and without DM.

### 4.1. SGLT2i and Renal Function

We demonstrated that eGFR was negatively associated with a recurrent hospitalization for HF and no improvement in plasma BNP levels during SGLT2i therapy. A subanalysis of EMPEROR-Reduced trial consistently demonstrated that hospital readmission decreased in relation to a decline in baseline eGFR [7].

Given the dominant mechanism of SGLT2i to increase urinary glucose excretion in urine, it would be plausible that the benefit of SGLT2i to prevent worsening HF diminishes with the progression of chronic kidney disease [8]. On the contrary, SGLT2i has a reno-protective effect particularly for those with chronic kidney disease with proteinuria. SGLT2i has pleiotropic effects independently of renal function [9,10]. Given all together, early administration of SGLT2i before deterioration of renal function would be highly recommended to prevent HF rehospitalization. Specific attention and careful monitoring for worsening HF would be encouraged particularly when SGLT2i is initiated in patients with worsening chronic kidney disease.

### 4.2. HFrEF, HFmrEF, and HFpEF

A pooled analysis using both the EMPEROR-Reduced and EMPEROR-Preserved trials showed that empagliflozin reduced the risk of hospitalization for HF irrespective of the ejection fraction levels [11]. However, the impact of ejection fraction on the clinical outcomes remains uncertain. This is the first study that demonstrated the differences in recurrent HF hospitalization rates between three EF groups treated with SGLT2i.

Importantly, there were obvious differences in the HF therapies, including beta-blockers, mineralocorticoid receptor antagonists, and diuretics, among three EF groups in this study (Table A2). Furthermore, our findings do not suggest that SGLT2i is less effective in HFrEF than in HFmrEF/HFpEF, because HFrEF generally have a higher hospitalization rate for HF than HFmrEF/HFpEF [12].

### 4.3. DM and Non-DM

In the DM cohort, there was a significant decrease in BNP immediately following the initiation of SGLT2i. We previously reported that urine volume was greater in patients with uncontrolled hyperglycemia compared with those with controlled hyperglycemia immediately after the initiation of SGLT2i, probably due to diuretic response and blood pressure reduction in the early phase [13].

In this study, recurrent HF hospitalization rate was comparable between patients with DM and those without DM. Large-scale trials consistently demonstrated greater risk reduction by SGLT2i regardless of the presence or absence of DM [1,2,3], except for a subanalysis of EMPEROR-Reduced trial, in which the impact of empagliflozin was neutral among those without DM [14]. Patients with DM might have favorable clinical benefit by SGLT2i during an acute phase, whereas its benefit might be compensated by relatively worse long-term prognosis of DM cohort [15]

### 4.4. Limitations

This study has the following limitations. First, this was a retrospective observational study conducted at a single center, and the sample size was small. Given the low event number, the number of potential confounders included in the multivariate analyses was restricted. Similarly, secondary outcome was only available for limited patients. Second, we should be careful to interpret the impact of SGLT2i in participants hospitalized for ADHF. In this study, treatment was given to compensate for HF, and other HF medications were also adjusted during index hospitalization. Furthermore, although all patients had NYHA class III/IV symptoms upon admission, we had not been able to accurately determine the NYHA classification of hospitalized patients at the time of their discharge. Also, it is challenging to conclude the effect of SGLT2i alone due to the lack of control group. Third, intergroup comparison has selection bias with different background characteristics between two groups. Notably, there were differences in the HF therapies among HFrEF, HFmrEF, and HFpEF patients. Also, there were differences in the antidiabetic therapies including types of SGLT2i between DM and non-DM patients. Fourth, this study did not include echocardiography follow-up, which is essential to determine the clinical course of chronic HF, especially in the subgroup with HFrEF and HFmrEF. Lastly, the multiple types of SGLT2i were used in the present study. It remains unclear whether such beneficial effects are consistent across individual SGLT2i. However, a recent retrospective cohort study suggested that there was no significant difference in the risk of cardiovascular events including HF among patients taking dapagliflozin, canagliflozin, and empagliflozin [16].

## 5. Conclusions

Among patients who received SGLT2i therapy during the hospitalization for ADHF, eGFR was associated with a recurrent hospitalization for HF following the index discharge. Early initiation of SGLT2i prior to the progression of chronic kidney disease may be recommended for patients admitted for ADHF.

## Figures and Tables

**Figure 1 jcm-11-05027-f001:**
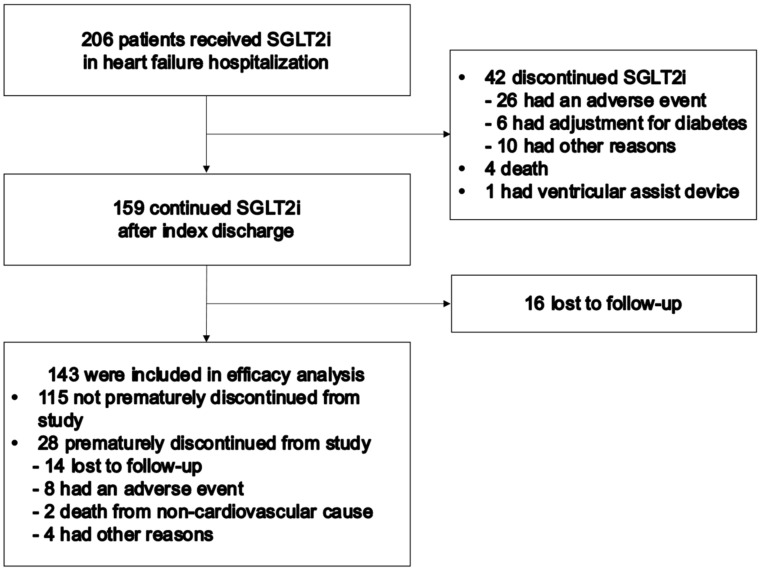
Enrollment and follow-up. SGLT2i, sodium-glucose cotransporter 2 inhibitor.

**Figure 2 jcm-11-05027-f002:**
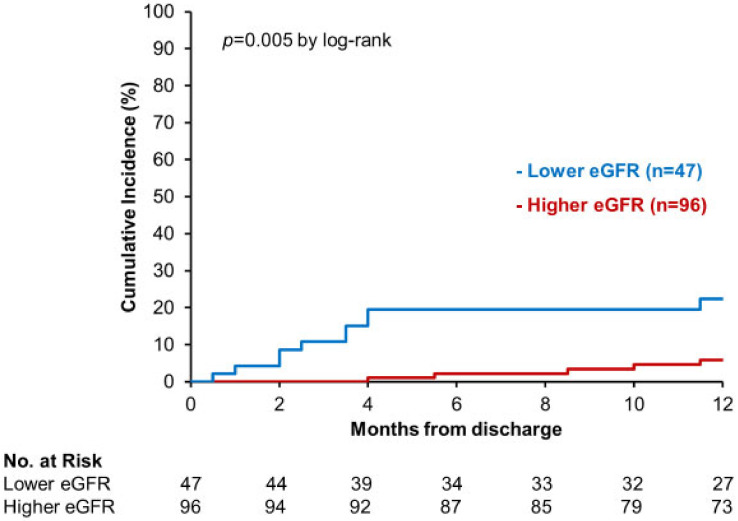
Recurrent hospitalization for heart failure in patients stratified by baseline estimated glomerular filtration rate (eGFR).

**Figure 3 jcm-11-05027-f003:**
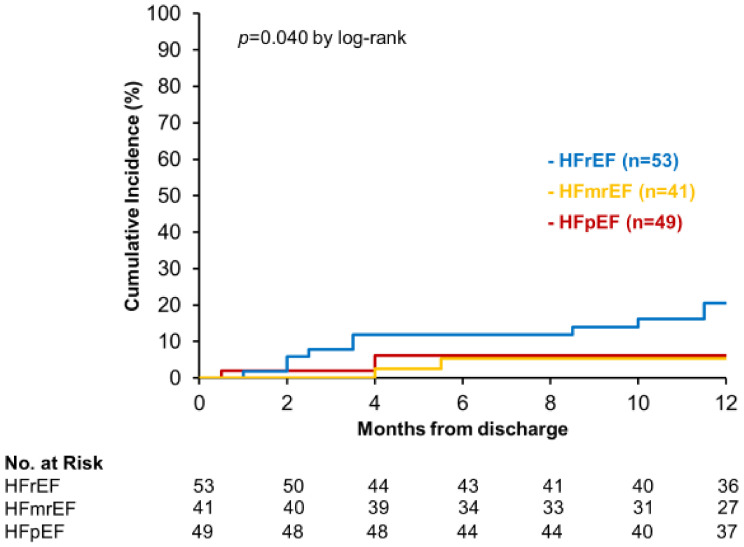
Recurrent hospitalization for heart failure in patients with heart failure stratified by LVEF category.

**Figure 4 jcm-11-05027-f004:**
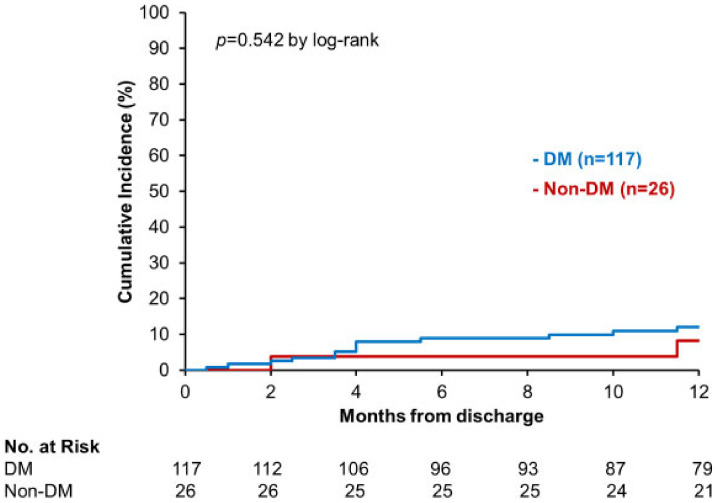
Recurrent hospitalization for heart failure in DM/non-DM patients.

**Figure 5 jcm-11-05027-f005:**
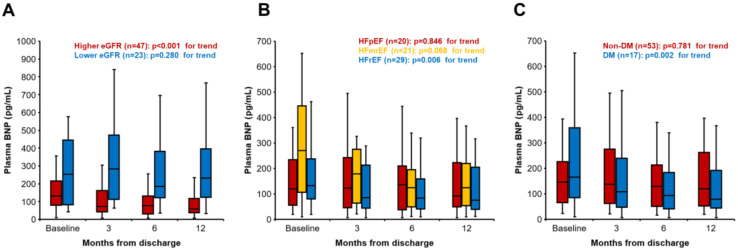
Changes in plasma B-type natriuretic peptide (BNP) level during the observational period. (**A**) Stratified by baseline estimated glomerular filtration rate (eGFR). (**B**) HFrEF/HFmrEF/HFpEF cohort. (**C**) DM/non-DM cohort. Variables were expressed as median and interquartile.

**Table 1 jcm-11-05027-t001:** Baseline characteristics.

Age, years	73 (65–81)
Male, *n* (%)	92 (64)
Body weight, kg	57.8 (48.5–68.3)
Body mass index, kg/m^2^	22.7 (19.8–24.9)
Systolic blood pressure, mmHg	107 (96–118)
Heart rate, beats per minutes	70 (63–878)
Ischemic etiology, *n* (%)	59 (41)
Atrial fibrillation, *n* (%)	42 (29)
Implantable cardioverter-defibrillator, *n* (%)	19 (13)
Cardio resynchronization therapy, *n* (%)	14 (10)
Left ventricular ejection fraction, %	44.0 (31.0–57.0)
Value of <40% (HFrEF), *n* (%)	53 (37)
Value of 40–49% (HFmrEF), *n* (%)	41 (29)
Value of ≥50% (HFpEF), *n* (%)	49 (34)
Diabetes mellitus, *n* (%)	117 (82)
HbA1c, %	6.8 (6.5–7.7)
Fasting blood sugar, mg/dL	110 (97–130)
Hemoglobin, g/dL	12.7 (11.2–14.1)
Hematocrit, %	38.0 (34.1–41.6)
Serum albumin, g/dL	3.6 (3.4–3.9)
Serum sodium, mEq/L	138 (136–140)
Serum potassium, mEq/L	4.4 (4.1–4.7)
eGFR, mL/minute/1.73 m^2^	50.5 (36.9–64.2)
Plasma BNP, pg/mL	142 (63–316)
Heart failure therapies	
Beta-blockers, *n* (%)	127 (89)
ACEI/ARB/ARNI, *n* (%)	131 (92)
Loop diuretics, *n* (%)	93 (65)
MRA, *n* (%)	96 (67)
Thiazides, *n* (%)	3 (2)
Antidiabetic agents	
Sulfonylureas, *n* (%)	6 (4)
DPP-4i, *n* (%)	62 (43)
Biguanides, *n* (%)	20 (14)
Insulin, *n* (%)	13 (9)
Sodium-glucose cotransporter 2 inhibitors	
Canagliflozin, *n* (%)	40 (28)
Dapagliflozin, *n* (%)	62 (43)
Empagliflozin, *n* (%)	41 (29)

HFrEF, heart failure with reduced ejection fraction; HbA1c, glycated hemoglobin; eGFR, estimated glomerular filtration rate; BNP, b-type natriuretic peptide; NT-proBNP, N-terminal pro-b-type natriuretic peptide; ACEI, angiotensin converting enzyme inhibitors; ARB, angiotensin receptor blockers; ARNI, angiotensin receptor-neprilysin inhibitors; MRA, mineralocorticoid receptor antagonists; DPP-4i, dipeptidyl peptidase-4 inhibitors.

**Table 2 jcm-11-05027-t002:** Variables associated with recurrent hospitalization for heart failure.

	All Patients (*n* = 143)					
	Univariable Analysis			Multivariable Analysis		
Variables	Hazard Ratio	95% CI	*p* Value	Hazard Ratio	95% CI	*p* Value
Age, years	1.03	0.99–1.09	0.198			
Male, yes	1.06	0.36–3.09	0.922			
Body mass index, kg/m^2^	0.84	0.72–0.98	0.030 *	0.88	0.71–1.05	0.206
Systolic blood pressure, mmHg	0.97	0.94–1.00	0.087			
Heart rate, bpm	1.02	0.97–1.06	0.436			
Ischemic etiology, yes	1.26	0.46–3.48	0.654			
Atrial fibrillation, yes	0.87	0.27–2.74	0.813			
HFrEF, yes	3.64	1.24–10.64	0.019 *	2.02	0.61–6.65	0.247
Diabetes mellitus, yes	1.58	0.36–7.01	0.547			
Fasting blood sugar, mg/dL	0.99	0.96–1.00	0.178			
Hematocrit, %	0.90	0.80–1.00	0.073			
Serum albumin, g/dL	0.28	0.08–0.93	0.039 *	0.60	0.16–2.47	0.469
Serum sodium, mEq/L	0.99	0.88–1.15	0.911			
Serum potassium, mEq/L	0.21	0.06–0.74	0.018 *	0.32	0.09–1.01	0.059
eGFR, mL/min/1.73 m^2^	0.95	0.92–0.98	0.003 *	0.94	0.90–0.98	0.007 *
ln BNP	2.57	1.42–4.90	0.003 *	1.50	0.80–3.05	0.233
Beta-blockers, yes	NA	NA	0.999			
ACEI/ARB/ARNI, yes	0.56	0.13–2.50	0.450			
Loop diuretics, yes	3.81	0.86–16.89	0.078			
MRA, yes	1.87	0.53–6.63	0.332			
Thiazides, yes	NA	NA	0.999			
Sulfonylureas, yes	NA	NA	0.999			
DPP-4i, yes	2.10	0.75–5.89	0.161			
Biguanides, yes	1.55	0.44–5.49	0.497			
Insulin, yes	0.73	0.10–5.52	0.757			

HFrEF, heart failure with reduced ejection fraction (ejection fraction < 40%); LVEF, left ventricular ejection fraction; HbA1c, glycated hemoglobin; eGFR, estimated glomerular filtration rate; BNP, b-type natriuretic peptide; ACEI, angiotensin converting enzyme inhibitors; ARB, angiotensin receptor blockers; ARNI, angiotensin receptor-neprilysin inhibitors; MRA, mineralocorticoid receptor antagonists; NA, not applicable. * *p* < 0.050.

## Data Availability

The data presented in this study are available on request from the corresponding author. The data are not publicly available due to privacy.

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
