# Peer review of "Factors Associated with Recurrent Heart Failure during Incorporating SGLT2 Inhibitors in Patients Hospitalized for Acute Decompensated Heart Failure"

_jcm, 2022, doi:10.3390/jcm11175027_

Round 1

Reviewer 1 Report

Interesting study although its findings may have to be considered within limits given its limitations, as you clearly stated.

Abstract

Line 12: replace “associating” with “associated”

Line 16: replace “associating” with “associated”

Line 50: replace “associating” with “associated”

Introduction

Line 46: replace “associating” with “associated”

Materials and Methods

Line 56: rearrange as follows “This study involved patients who had been admitted for ADHF...”

Line 75-77: a cut off of 40% for LVEF may not be appropriate to define HFpEF

Results

Table 2: replace “Variables associating” with “Variables associated”

Line 135-136: The effect of SGLT2i on the primary outcome was consist between patients with DM and those without DM. Do you mean “was consistent”? It would be appropriate to rephrase and explain this aspect better.

Discussion

Line 150: replace “associating” with “associated”

Line 159: replace with “...hospital re-admission decreased in relation to a decline in baseline eGFR [7].”

Line 168: replace with “worsening chronic kidney disease”

Line 207-209: it would be most appropriate to rephrase the whole period because it is not clear. Do you mean that the benefit on HF reflects the NYHA Class?

Conclusion

Given the limitations of the study, it would be more appropriate to say “Early initiation of SGLT2i prior to the progression of chronic kidney disease may be recommended in patients admitted for ADHF.”

Reviewer 2 Report

I read with interest the paper of and it was generally well presented. 

Just some minor comments for the Authors: 

-In the results the Authors report that DM was noted in 26 patients (page 4, line 117), while in the table they report 117 patients. Please correct it.

-I am quite surprised for the high number of patients treated with empaglifozin in the acute phase in the LVEF group above 40% since data of Emperor Preserved are quite recent. Was the prescription reserved only to patients with DM?

-Last point: do you have data about LVEF of patients who were furtherly hospitalized for heart failure? It was reported that SGLT2 inhibitors might in part contribute to left ventricular reverse remodelling and it would be nice to see if patients with HFrEF who were hospitalized for HF had an improvement of LVEF since baseline. It was also recently reported that the LVEF trajectory is an important predictor of rehospitalization in patients with HF (Manca P, Stolfo D, Merlo M, Gregorio C, Cannatà A, Ramani F, Nuzzi V, Lund LH, Savarese G, Sinagra G. Transient versus persistent improved ejection fraction in non-ischaemic dilated cardiomyopathy. Eur J Heart Fail. 2022 Jul;24(7):1171-1179. doi: 10.1002/ejhf.2512). The authors could discuss this topic more extensively. 

Reviewer 3 Report

This article contains a serious terminological error. According to the current Japanese and Western guidelines, the LVEF-based classification of HF proposes three classes, not two as mentioned by the authors. It seems that the authors do not accept the modern classification of HF. They completely ignore the HFmrEF, determining the HFpEF at an ejection fraction >40%. That is not correct.

Throughout the article, the authors describe two groups of HFrEF and HFpEF and define only the first group (LVEF <40%). The wrong definition of HFpEF as LVEF > 40% does not appear until the end of the article (suppl. table A1 and A2, p.231,232). The authors should either correctly define the  groups according to guides or determine the groups of participants in the study depending on LVEF at their own discretion. For example: as <40% and >40% without misusing the definitions of the modern classification of heart failure.

I agree that no classification is perfect.  This is why heart failures with LVEF 49% and 51%, despite belonging to different classes, are hard to distinguish clinically. However, the terminology used in a scientific paper must be correct. In addition, the authors refer and compare their findings to recent studies on the use of SGLT2 inhibitors, in all of which the participants were divided by LVEF into three classes with preserved, mid-range, and reduced ejection fraction. 

In conclusion, the manuscript is clear, relevant to the field, and interesting for a wide readership. However, it is necessary to correct the terminology inaccuracies mentioned above.

Round 2

Reviewer 1 Report

No further comments